# The Roles of Two CNG Channels in the Regulation of Ascidian Sperm Chemotaxis

**DOI:** 10.3390/ijms23031648

**Published:** 2022-01-31

**Authors:** Kogiku Shiba, Kazuo Inaba

**Affiliations:** Shimoda Marine Research Center, University of Tsukuba, 5-10-1, Shimoda 415-0025, Japan; kinaba@shimoda.tsukuba.ac.jp

**Keywords:** calcium, sperm chemotaxis, fertilization, cAMP, cGMP

## Abstract

Spermatozoa sense and respond to their environmental signals to ensure fertilization success. Reception and transduction of signals are reflected rapidly in sperm flagellar waveforms and swimming behavior. In the ascidian *Ciona intestinalis* (type A; also called *C. robusta*), an egg-derived sulfated steroid called SAAF (sperm activating and attracting factor), induces both sperm motility activation and chemotaxis. Two types of CNG (cyclic nucleotide-gated) channels, Ci-tetra KCNG (tetrameric, cyclic nucleotide-gated, K^+^-selective) and Ci-HCN (hyperpolarization-activated and cyclic nucleotide-gated), are highly expressed in *Ciona* testis from the comprehensive gene expression analysis. To elucidate the sperm signaling pathway to regulate flagellar motility, we focus on the role of CNG channels. In this study, the immunochemical analysis revealed that both CNG channels are expressed in *Ciona* sperm and localized to sperm flagella. Sperm motility analysis and Ca^2+^ imaging during chemotaxis showed that CNG channel inhibition affected the changes in flagellar waveforms and Ca^2+^ efflux needed for the chemotactic turn. These results suggest that CNG channels in *Ciona* sperm play a vital role in regulating sperm motility and intracellular Ca^2+^ regulation during chemotaxis.

## 1. Introduction

The changes in sperm motility before fertilization are a conserved mechanism among most organisms for fertilization success in high efficiency. Spermatozoa sense and respond to environmental signals such as several ion concentrations and chemical ligands derived from the egg or female reproductive organs [1,2]. The signal receptions and transduction rapidly lead to molecular motor activation, resulting in sperm motility activation and flagellar regulation to control the swimming path. The regulation of intracellular Ca^2+^ concentrations, pH, and ion channels and exchangers mediating membrane potential is a key factor for sperm signaling [3,4,5].

We have studied the regulatory mechanism of sperm flagellar motility in the ascidian, *Ciona intestinalis* (type A; also called *C. robusta*) to understand the molecular basis for the changes in sperm motility necessary for fertilization [6,7]. *Ciona* is a hermaphroditic marine invertebrate that simultaneously spawns sperm and eggs to fertilize but shows self-sterility. *Ciona* sperm present clear chemotaxis toward the egg, and the chemoattractant is identified from the egg called SAAF, a sperm activating and attracting factor. SAAF is a sulfate-conjugated hydroxysteroid [8] and interacts with the plasma membrane Ca^2+^-ATPase (PMCA), suggesting that PMCA is a potent candidate for the SAAF receptor [9].

In the sea urchin, a guanylyl cyclase (GC) is identified as the receptor for sperm chemotaxis, suggesting that the initial signaling pathways in sperm chemotaxis between ascidian (*Ciona*) and sea urchins are different. In this study, we discussed two types of CNG channels, Ci-tetra KCNG (tetrameric, cyclic nucleotide-gated, K^+^-selective) and Ci-HCN (hyperpolarization-activated and cyclic nucleotide-gated). These two channels have been reported to be vital for the signaling pathways in sea urchin sperm [10,11,12,13]. The Sp-tetra KCNG channel has been considered to induce cGMP-dependent membrane hyperpolarization and regulate Ca^2+^ dynamics in sea urchin sperm chemotaxis [12,14]. HCN channels (Sp-HCN1 and 2) contribute to a Na^+^ influx in a cAMP-dependent manner in sea urchin sperm [10,11].

We previously reported that these CNG channels are highly expressed in *Ciona* testis from the comprehensive gene expression analysis [15]. Here, we found high expression of both CNG channel proteins in *Ciona* spermatozoa by Western blotting and showed the localization to flagella by immunofluorescence analysis. Additionally, we analyzed sperm motility and Ca^2+^ dynamics in the presence of a CNG channel inhibitor. Our results indicate that CNG channels regulate sperm chemotaxis by controlling Ca^2+^ efflux. We also discuss the roles of these CNG channels and cyclic nucleotides on sperm motility regulation in marine invertebrates by comparing sea urchin and ascidians or cAMP and cGMP.

## 2. Results

### 2.1. Phylogenetic and Sequence Analysis

We previously found that genes for the two types of CNG channels, Ci-tetra KCNG (tetrameric, cyclic nucleotide-gated, K^+^-selective) and a Ci-HCN (hyperpolarization-activated and cyclic nucleotide-gated), are highly expressed in testis [15]. BLASTP search of these channels against the Ciona proteome found two further types of HCN channels (Ci-HCN2, KH.L96.86; Ci-HCN3, KH.C10.454) in addition to Ci-HCN1 (KH.C10.165) and three other CNG channels (Ci-CNG, Ci-CNG3, and Ci-CNG4) with similar sequence to human CNG orthologs. The phylogenetic tree analysis revealed that Ci-tetraKCNG is grouped into the same clade as vertebrate orthologs (Fishes) but not into the marine invertebrate clade such as those in sea urchin. Alternatively, Ci-HCN1, 2 and 3 are not grouped into the clade of vertebrates, sea urchins, or arthropods but into a separate clade closer to that of vertebrates (Figure 1).

The Ciona KCNG channel (Ci-tetraKCNG; KH.C7.121) is a single polypeptide with four KCNG domains, each of which contains six transmembrane segments (S1–6), a K^+^-selective ion pore, and a cyclic nucleotide-binding domain (CNBD) that connects to the adjacent transmembrane segment in a KCNG domain (Figure 2A). The predicted structure was quite similar to that of the sea urchin KCNG channel. Amino acid sequence alignments of the CNBD and the pore region showed high similarity between the Sp-tetraKCNG channel and the Ci-tetraKCNG channels (Figure 2B,C). Conversely, both Ciona HCN1 (Ci-HCN1) and HCN2 channels (Ci-HCN2) are single polypeptides containing six transmembrane segments (S1–6), an ion pore region, and a cyclic nucleotide-binding domain (CNBD) (Figure 3A). The Ciona HCN channels also showed high similarities to sea urchin HCN channels (Figure 3B,C).

### 2.2. Expression of Tetra-KCNG and HCN Genes in Ciona Testis

Since both HCN and tetraKCNG play key roles in regulating sperm motility in sea urchins [10,11,12,13], we examined the specificity in the expression of these CNG genes in Ciona tissues by reverse-transcriptase PCR (RT-PCR). The Ci-tetraKCNG, HCN1, and HCN2 expressions were testis-specific; however, Ci-HCN3 showed no expression either in testis or other tissues examined (Figure 4).

### 2.3. Immunoblotting and Immunofluorescent Microscopy

To elucidate the function of Ci-tetraKCNG and Ci-HCN in Ciona sperm, we prepared polyclonal antibodies against bacteria-expressed polypeptides of these channels. Western blotting showed that the antibody against Ci-tetraKCNG specifically recognized a protein band with a large molecular mass in the whole sperm proteins. The band’s size agreed well with the molecular mass (~271 kDa) expected from the amino acid sequence (Figure 5A). Conversely, the antibody against Ci-HCN2 recognized two bands in Triton X-100 soluble sperm proteins (Figure 5C). These bands were Ci-HCN1 and Ci-HCN2 because they share many regions showing highly identical amino acid sequences (Figure 3). The molecular mass of the two bands was estimated as 81.2 and 73.2 kDa, which agreed with those predicted for Ci-HCN2 (79.1 kDa) and Ci-HCN1 (69.7 kDa), respectively. Immunofluorescent microscopy clearly showed that both Ci-tetraKCNG and Ci-HCNs are localized along the entire length of the sperm flagellum (Figure 5B,D).

### 2.4. Effects of HCN Inhibitors on the Chemotaxis of Ciona Sperm

As already shown in sea urchin and zebrafish, tetraKCNG induces chemoattractant-dependent membrane hyperpolarization [13,14,16,17], which has been widely conserved in the sperm of marine invertebrates, including Ciona intestinalis [18]. In the sea urchin, membrane hyperpolarization activates the HCN channel to induce membrane depolarization, resulting in the Ca^2+^ influx through CatSper [19,20]. However, the role of HCN is unknown in Ciona sperm. To explore this, we examined the effect of a cAMP-gated cation channel inhibitor, ZD7288, which inhibits an HCN channel in sea urchin sperm [12].

Chemotactic behavior in the Ciona sperm consists of sharp turning movement and straight swimming, resulting in swimming toward the chemoattractant’s source [21]. The highly asymmetric flagellar waveform triggered by transient Ca^2+^ influx is needed for the turning movement [21]. The CNG inhibitor ZD7288 reduced the sharpness of turning by more than 10 μM (Figure 6A). Quantitative analysis showed that ZD7288 affected swimming velocity at higher concentrations but significantly lowered the linear equation chemotaxis index, LECI (Figure 6B), suggesting that it affects Ca^2+^ dynamics in the Ciona sperm during chemotaxis.

We next examined the changes of the intracellular Ca^2+^ concentration in Ciona sperm during chemotaxis. The transient Ca^2+^ increase and decrease during the chemotactic turn was observed in the control (Figure 7A, control). Alternatively, the Ciona sperm treated by ZD7288 showed excess Ca^2+^ influx and extended the period of high Ca^2+^ concentration (Figure 7A, ZD7288). The maximum value of the Ca^2+^ intensity and duration of Ca^2+^ burst was increased significantly by treatment of ZD7288 (Figure 7B,C). These results suggest that the HCN channel plays an important role in the Ca^2+^ signaling required for proper chemotactic movement.

## 3. Discussion

This study showed that two types of CNG channels, tetraKCNG and HCN, are highly expressed and localized along sperm flagella in the ascidian *Ciona intestinalis*. In sea urchin sperm, K^+^-dependent membrane hyperpolarization and following HCN-dependent depolarization via Na^+^ influx leads to Ca^2+^ influx through CatSper and plays a vital role in signal transduction for chemotaxis [22,23,24]. Thus, a similar mechanism is believed to control sperm chemotaxis in ascidian sperm. However, the receptor of a chemoattractant is a membrane-bound GC in sea urchin sperm [25], whereas that in ascidian sperm is shown to be a plasma membrane Ca^2+^-ATPase (PMCA) [9]. No ortholog of the sea urchin GC was found in testis-expressed sequence tags [15] or the testis proteome [26] in *Ciona intestinalis*, suggesting that cGMP-mediated signal transduction is not present in the process of ascidian sperm chemotaxis. Similar to sea urchins, it was recently discussed in corals that receptor-bound GC would be expressed in sperm and activates KCNG, resulting in hyperpolarization [26]. However, neither zebrafish nor mammals appear to use GC for membrane hyperpolarization [27,28]. Taken together, the molecular species and the ligand-mediated synthesis of cyclic nucleotides for the regulation of sperm chemotaxis might have largely changed in chordates during animal evolution.

High concentration of extracellular K^+^ completely inhibits sperm chemotaxis in *Ciona* [18], suggesting that K^+^ efflux through tetraKCNG and subsequent membrane hyperpolarization might be essential pathways in *Ciona* sperm chemotaxis. Dissociation of SAAF from the receptor, PMCA, is thought to terminate Ca^2+^ efflux and increase intracellular Ca^2+^ concentration when sperm swim away from the chemoattractant source [9]. An increase in Ca^2+^ activates adenylyl cyclase (AC) to produce cAMP, which would activate tetraKCNG and accelerate K^+^ efflux, resulting in membrane hyperpolarization. SAAF-induced membrane hyperpolarization by tetraKCNG is thought to stimulate the HCN channel. In this study, we showed that the HCN channel inhibitor induced excess Ca^2+^ influx and the suppression of sperm chemotactic behavior, suggesting that the HCN channel is involved in the Ca^2+^ efflux when sperm swim toward the attractant source.

We previously reported that two distinct types of AC are present in *Ciona* sperm [7]. The transmembrane adenylyl cyclase (tmAC) is involved in the activation of sperm motility, whereas soluble adenylyl cyclase (sAC) influences Ca^2+^-dependent chemotactic movement of sperm. The activation is stimulated by the phosphorylation of dynein subunits and other axonemal proteins by a cAMP-dependent protein kinase [6,26]. However, the mechanism of how sAC controls the chemotaxis in *Ciona* has not been elucidated. We showed here that HCNs are involved in intracellular Ca^2+^ dynamics essential for the chemotactic movement of sperm. Therefore, cAMP generated by sAC is likely to activate HCN channels for membrane depolarization, playing a role to keep intracellular pH and membrane potential as a pacemaker to control intracellular Ca^2+^ dynamics through binding of SAAF to PMCA [9] and/or Na^+^/Ca^2+^ exchanger [29].

The regulation of sperm motility by sAC and pH is also shown in lower animals, such as corals [28]. In mammals, the compartmentalization of two types of AC is observed: tmAC and sAC localized in sperm head and flagella, respectively [27]. Although we have not examined the localization of two types of AC in *Ciona* sperm, sAC plays key roles in flagella both in the increase in beat frequency and in the change in flagellar waveform during chemotaxis [7]. Thus, cAMP appears to control sperm motility in two opposite pathways, i.e., the activation of tetraKCNG in response to Ca^2+^ increase under higher SAAF conditions (toward the chemoattractant) and the activation of HCN in response to membrane hyperpolarization under lower SAAF conditions (away from the chemoattractant). It would be intriguing to know the mechanism by which sAC properly synthesizes and provides cAMP for the regulation of two CNG channels.

## 4. Materials and Methods

### 4.1. Materials

The ascidian *C. intestinalis* (type A; also called *C. robusta*) was collected from Onagawa Bay near the Onagawa Field Research Center, Tohoku University or obtained from the National BioResource Project for *Ciona* (http://marinebio.nbrp.jp/; accessed on 30 December 2021). Animals were kept in aquaria under constant light for the accumulation of gametes without spontaneous spawning. Semen samples were collected by dissecting the sperm duct and kept on ice until use.

### 4.2. Chemical and Solutions

Artificial seawater (ASW) was composed of 462.01 mM NaCl, 9.39 mM KCl, 10.81 mM CaCl_2_, 48.27 mM MgCl_2_ and 10 mM Hepes-NaOH (pH 8.0). ZD7288 (Tocris Bioscience, Bristol, UK) was dissolved in water and added to an appropriate concentration in ASW. Other reagents were of analytical grades. SAAF was synthesized as described previously [30,31].

### 4.3. Sequence Analysis

The alignment and phylogenetic tree were constructed by MUSCLE with the aid of MEGA X [32].

### 4.4. Reverse-Transcriptase (RT) PCR

Total RNA was extracted from the endostyle, gill, testis and ovary of *C. intestinalis* by using a QIAGEN RNeasy mini kit (QIAGEN, Hilden, Germany). cDNA was synthesized from the total RNA of each tissue by using SuperScript III reverse transcriptase (Invitrogen, Carlsbad, CA, USA). PCR was performed with Platinum^®^ Taq (Invitrogen) and specific primers as follows: Ci-tetraKCNG, 5′-AAGTCCTGACTACCTCCTCATCGGT-3′(forward) and 5′-GTTCATGACCATGTCGAGCGCAAA-3′(reverse); Ci-HCN1, 5′-CCCCTTCACAAAATGGTCCCGAAA-3′(forward) and 5′-GGCTGGTGCTTTTGAACAAAGACC-3′(reverse); Ci-HCN2, 5′-AGATGGCGCTTGCTTTCGCT-3′(forward) and 5′-TATGCTGGGCGGACTTGATCGT-3′(reverse); Ci-HCN3, 5′-GTTCATTGGTCATGCCACGGCATT-3′(forward) and 5′-CAAAGACACAGCGCCACGAACT-3′(reverse); Ci-β-Actin, 5′-GTGCTTTCATTGTACGCTTCTGGTC-3′(forward) and 5′-CGGCGATTCCAGGGAACATAG-3′(reverse). Amplifications were carried out for 30 cycles (94 °C for 30 s, 55 °C for 30 s and 72 °C for 1 min).

### 4.5. Preparation of Antibodies and Immunoblotting

A polyclonal antibody against Ci-tetraKCNG and Ci-HCN2 was raised in mouse as previously performed (Mizuno et al., 2009). The following PCR primers were used for amplification of the open reading frame for Ci-tetraKCNG (5′-GCGCGGATCCATGCTTACGAACATTGA-3′ (sense) and 5′-GCGCGAATTCTTAGTTGGCATTCTGTGC-3′ (antisense)); Ci-HCN2 (5′-ATGGGTCGCGGATCCGTAAAGGAATACATG-3′ (sense) and 5′-ACGGAGCTCGAATTC AGGCGTGCTTATTTC-3′ (antisense)). The PCR product for Ci-tetraKCNG was subcloned into a pET32a vector and transfected into *Escherichia coli* AD494. The PCR product for Ci-HCN2 was subcloned into a pET28a vector and transfected into *Escherichia coli* BL21(DE3). Protein expression was induced by 0.5 mM IPTG (isopropyl β-D-thiogalactoside). The recombinant proteins were purified using Chelating Sepharose Fast Flow (GE healthcare). Semen was suspended in 100 volumes of ASW and solubilized in a sample buffer; 62.4 mM Tris-HCl, 2% SDS, 4% glycerol, 0.004% bromophenol-blue, pH 6.8 and boiled at 37 °C for 20 min. To prepare triton soluble sperm protein, semen was suspended in 25 volumes of Ca^2+^-free SW and centrifuged at 2000× *g* for 5 min at 4 °C. The resulting pellet was suspended in an extraction medium containing 0.15 M KCl, 2 mM MgCl_2_, 0.5 mM EGTA, 0.1% Triton X-100 and 10 mM Tris-HCl at pH 8.0. The suspension was centrifuged at 17,900× *g* for 5 min at 4 °C, and the supernatant was mixed with the sample buffer and boiled at 37 °C for 20 min. Proteins were separated by SDS-PAGE and transferred to polyvinylidene difluoride membranes. Membranes were treated with 7.5% skim milk in PBST (PBS containing 0.1% Tween 20) to prevent non-specific protein binding. Blots were incubated with the anti-Ci-tetraKCNG (1:5000) or anti-Ci-HCN2 (1:2000) primary antibodies for 1 h at room temperature. After washing with PBST three times, blots were incubated with HRP-conjugated secondary antibodies at 1:10,000 for 30 min at room temperature. After washing with PBST three times. The immunoreactive bands were detected using ECL-prime (GE Healthcare, Chalfont St Giles, UK). Signals were detected using the LAS-4000 mini imager (Fujifilm, Tokyo, Japan).

### 4.6. Immunofluorescence Microscopy

Immunofluorescence microscopy was performed as in a previous study [6] with slight modifications. Semen was suspended in 50 volumes of ASW and attached on slides pre-coated with 1 mg/mL poly-l-lysine. After incubation for 10 min, the sperm were fixed by methanol at −30 °C for 10 min and rehydrated by excess volume of PBS. After incubation with PBS containing 0.1% TritonX-100 for 5 min, the slides were incubated with blocking buffer (10% goat serum in PBS) for 1h in a moist chamber, followed by incubation with mouse primary antibody against Ci-tetraKCNG or Ci-HCN2, or nonimmune mouse serum at 1:200 dilution in the blocking buffer for 1 h. After washing with PBS, the slides were incubated with goat anti-mouse IgG (Alexa 488; Molecular Probes, Eugene, OR, USA) (1:1000) and Monoclonal Anti-β-Tubulin−Cy3 antibody (C4585; Sigma-Aldrich, St. Louis, MO, USA) (1:200) for 1 h. After washing with PBS, samples were briefly treated with 4′,6-diamidino-2-phenylindole dichloride (DAPI) for DNA staining and mounted in 50% glycerol in PBS. The slides were observed under a fluorescence microscope (BX53; Olympus, Tokyo, Japan) with 100× objective (UPlanApo, Olympus, Tokyo, Japan).

### 4.7. Assessment of Sperm Chemotaxis

Sperm chemotaxis was examined as described previously [21]. Briefly, semen was suspended in 2000 volumes of ASW or ASW with inhibitor. After incubation for 3 min, the sperm suspension was placed in the observation chamber, and sperm movement around the micropipette tip containing 1 μM SAAF was observed under a phase contrast microscope (BX51, Olympus, Tokyo, Japan) with a 10× objective (UPlan FL N, Olympus, Tokyo, Japan) and recorded by a CCD camera (UI-3360CP-C-HQ, iDS, Obersulm, Germany) at 29.96 frames per second. The position of the sperm head was analyzed with Bohboh software (BohbohSoft, Tokyo, Japan). The parameter indicating the strength of sperm-attracting activity, LECI (linear equation chemotaxis index) was calculated from the distance between the capillary tip and sperm position, as described previously [8].

### 4.8. Imaging Analysis of Intracellular Ca^2+^


For Ca^2+^ imaging, Fluo-8H AM (AAT Bioquest, Sunnyvale, CA, USA) was used as a fluorescent probe. The dye-loaded sperm was prepared as described previously (Shiba et al., 2009). Fluorescent images of the sperm were observed by a microscope (IX71, Olympus, Tokyo, Japan), and captured on a PC connected to a digital CCD camera (ImagEM, C9100-13; Hamamatsu Photonics, Hamamatsu, Japan) at 32.7 frames/s using Aquacosmos (Hamamatsu Photonics). For fluorescence illumination, a stroboscopic lighting system with a power LED was used as described. Fluorescent signal intensity and sperm swimming trajectories were also analyzed using the Bohboh software.

### 4.9. Statistical Analysis

All experiments were repeated at least three times with three different animals. Data are expressed as means ± SE. Statistical significance was calculated using Dunnett’s test, Steel’s multiple comparison test and Student’s *t*-test; *p* < 0.05 was considered significant. The data were analyzed by using the R script.

## 5. Conclusions

Two types of CNG channels, tetra KCNG and HCN, are localized in the sperm tail in the ascidian *Ciona intestinalis*. A CNG channel inhibitor causes excess Ca^2+^ influx and extension of Ca^2+^ increase duration in sperm during chemotactic turn, resulting in the failure of chemotactic behavior. Thus, CNG channels in *Ciona* sperm play a vital role in regulating sperm motility and intracellular Ca^2+^ regulation during chemotaxis.

## Figures and Tables

**Figure 1 ijms-23-01648-f001:**
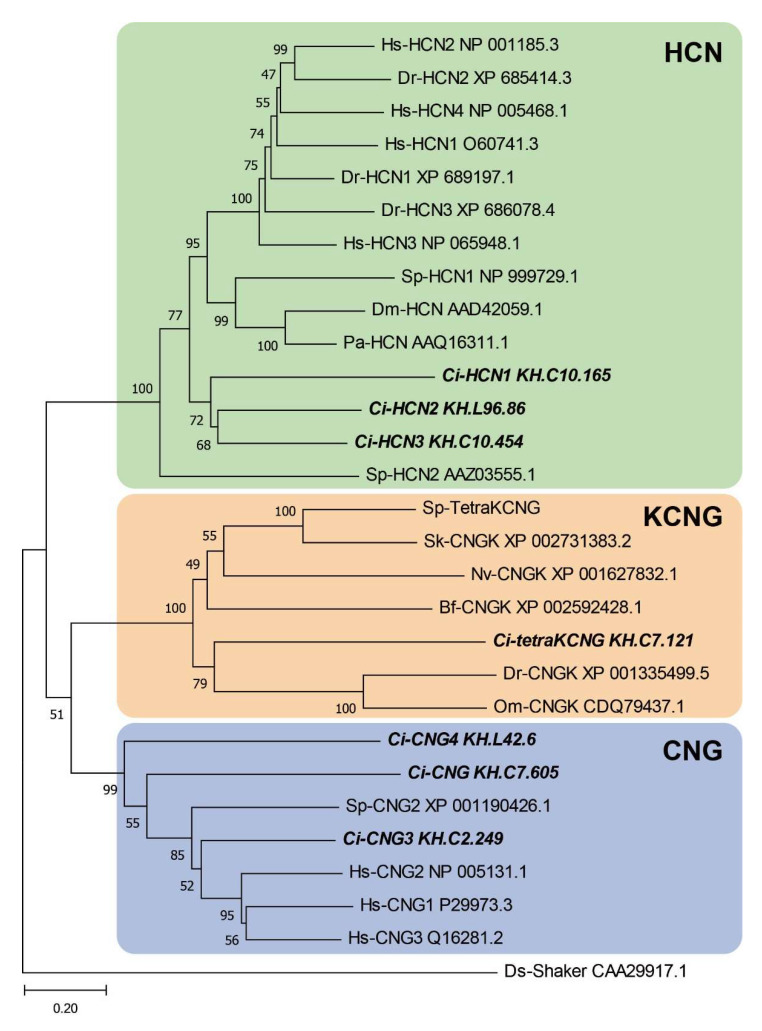
Phylogenetic analysis of cyclic nucleotide-gated channels from *Ciona*, sea urchins and other animals. The following ion channel sequences were used for the alignment: the HCN channel from *Ciona* (Ci-HCN1, 2, 3), sea urchin (Sp-HCN1, 2), human (Hs-HCN1, 2, 3, 4), zebrafish (Dr-HCN1, 2, 3), fruit fry (Dm-HCN), spiny lobster (Pa-HCN); the CNG channel from *Ciona* (Ci-CNG, 3, 4), sea urchin (Sp-CNG2), human (Hs-CNG1, 2, 3); CNGK channels from *Ciona* (Ci-tetraKCNG), sea urchin (Sp-tetraKCNG), acorn worm (Sk-CNGK), amphioxus (Bf-CNGK), starlet sea anemone (NvCNGK), zebrafish (Dr-CNGK) and rainbow trout (OmCNGK). The Drosophila shaker channel (Ds-Shaker) was used as an outgroup. The value on each branch represents the number of times that a node was supported in 100 bootstrap pseudoreplications.

**Figure 2 ijms-23-01648-f002:**
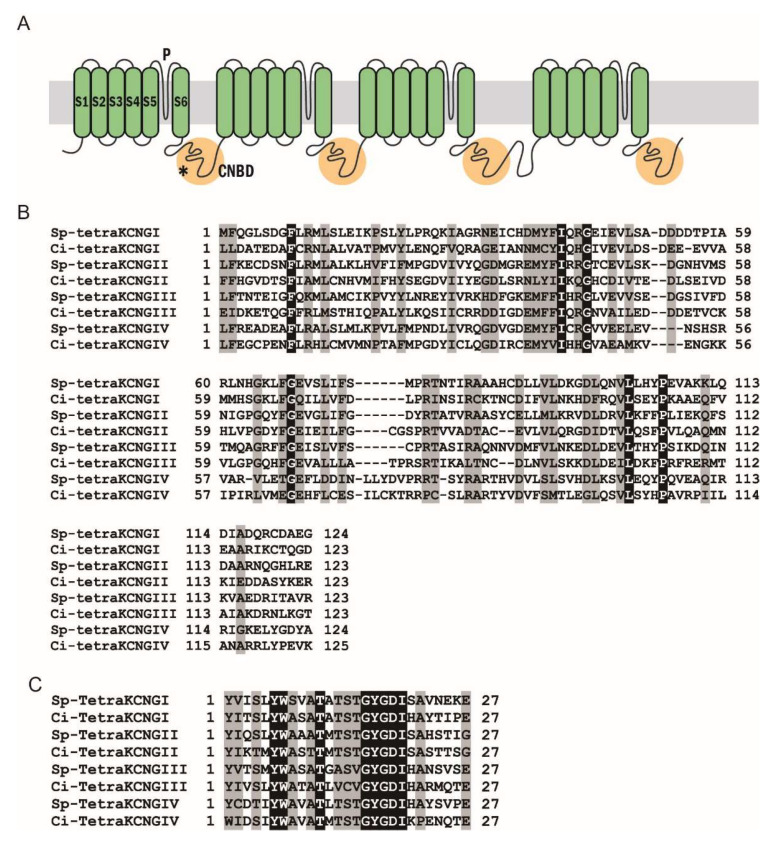
(**A**) Schematic representation of the tetraKCNG channel. * The asterisk shows the region used for the antigen to raise a polyclonal antibody. (**B**) Amino acid sequence alignment of the cyclic nucleotide-binding domain (CNBD) from the Sp-tetraKCNG channel and the Ci-tetraKCNG channel. The tetraKCNG channel identified in *Ciona* is very similar to the tetraKCNG channel in sea urchin. (**C**) Amino acid sequence alignment of the pore region.

**Figure 3 ijms-23-01648-f003:**
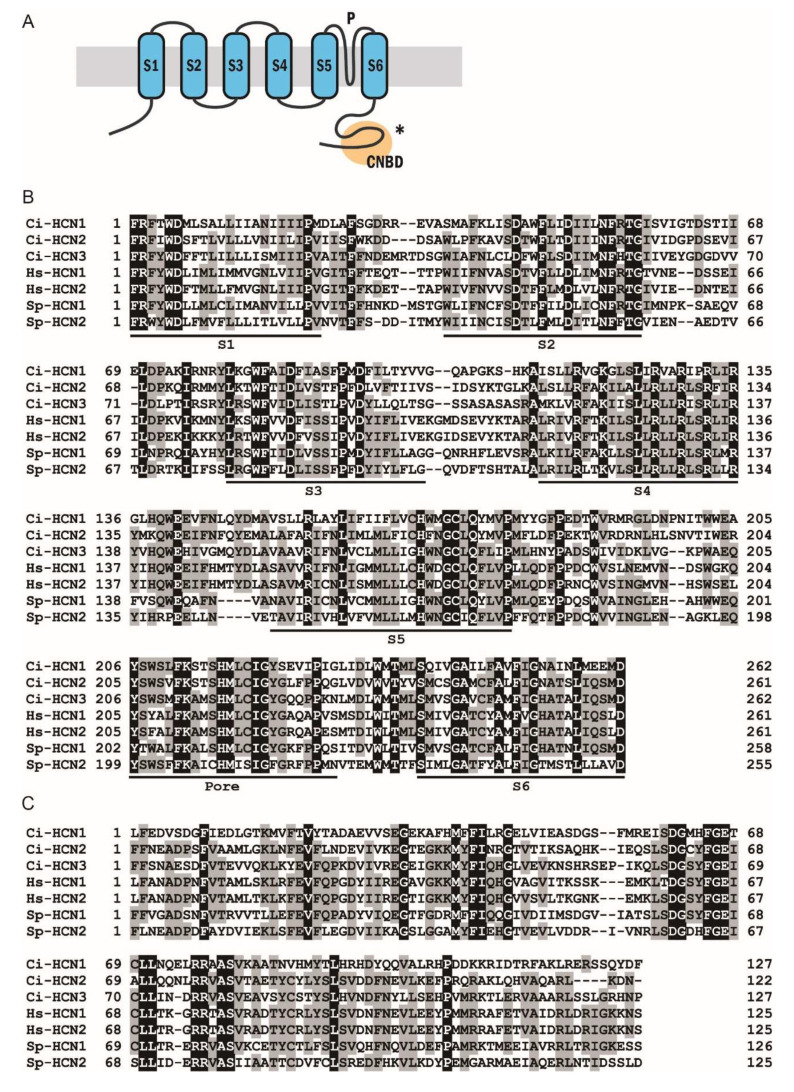
(**A**) Schematic representation of the HCN channel. * The asterisk shows the region used for antigen to raise a polyclonal antibody. (**B**) Amino acid sequence alignment of the transmembrane segments S1–6 and the pore region from the Sp-HCN1, 2, Ci-HCN1 and 2. (**C**) Amino acid sequence alignment of the cyclic nucleotide-binding domain (CNBD) from the Sp-HCN1, 2, Ci-HCN1 and 2.

**Figure 4 ijms-23-01648-f004:**
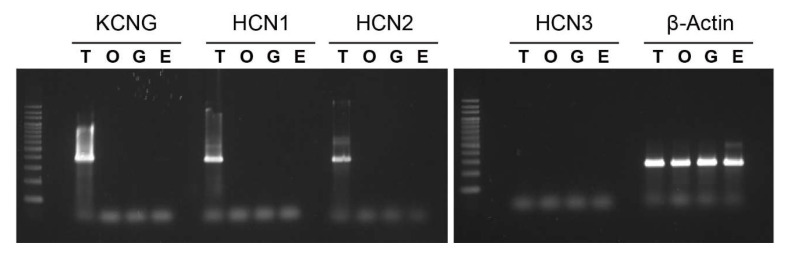
Tissue expression patterns of the CNG genes in *Ciona* tissues. RT-PCR analysis of mRNA from several adult *Ciona* tissues shows that the expression of KCNG, HCN1 and 2 is testis-specific. β-Actin was used as an internal control. E, endostyle; G, gill; T, testis; O, ovary.

**Figure 5 ijms-23-01648-f005:**
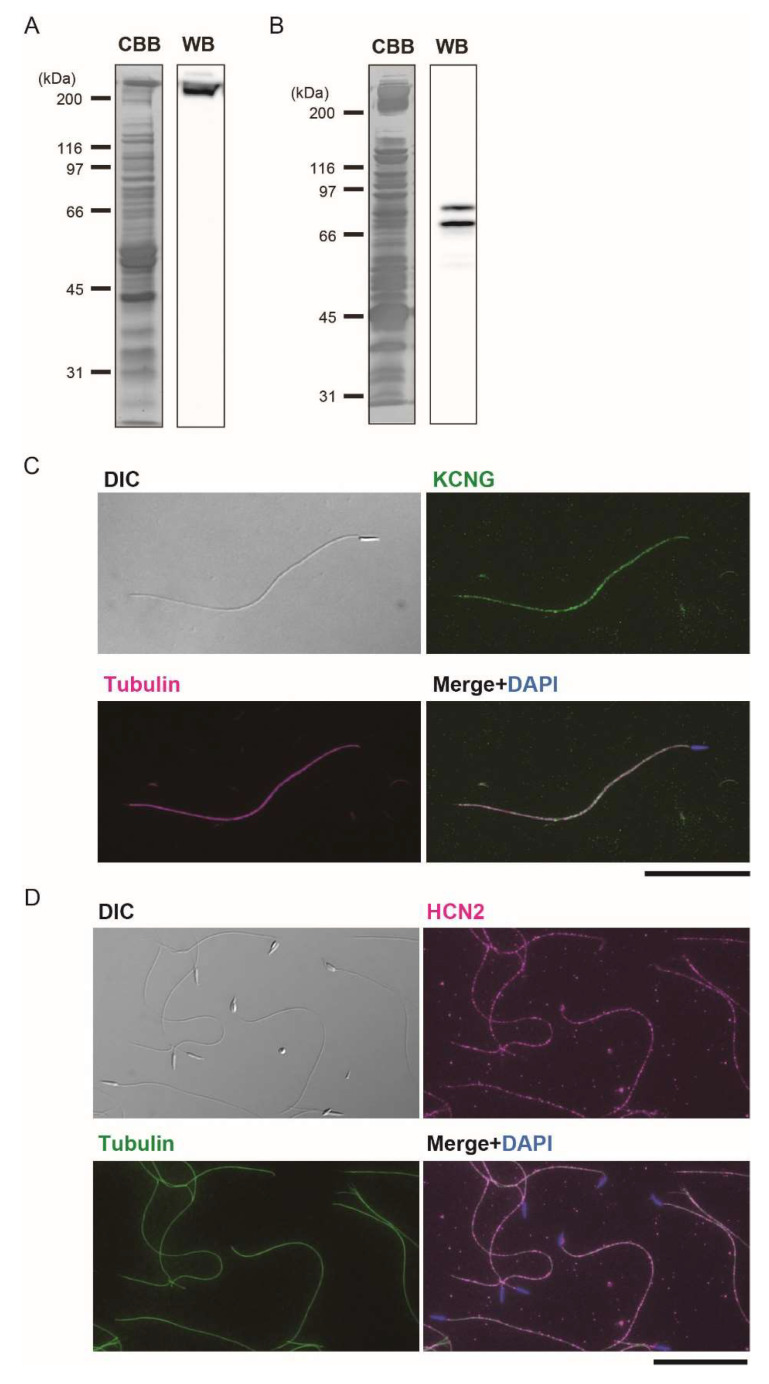
Western blot and immunolocalization analysis of the Ci-tetraKCNG and Ci-HCN2 in *Ciona* sperm. (**A**) Western blot of whole sperm protein with the antibody against the Ci-tetraKCNG. CBB-stained pattern of whole sperm proteins (Left) and corresponding immunoblots (Right) are shown. (**B**) Immunolocalization with the antibody against the Ci-tetraKCNG. The differential interference contrast (DIC) image, Ci-tetraKCNG (green), β-tubulin-Cy3 (magenta), and the merged image with DAPI (blue) are shown. Scale bar: 20 µm. (**C**) Western blot of Triton X-100 soluble sperm protein with the antibody against the Ci-HCN2. CBB-stained pattern of whole sperm proteins (Left) and corresponding immunoblot (Right) are shown. (**D**) Immunolocalization with the antibody against the HCN2. The differential interference contrast (DIC) image, Ci-HCN2 (magenta), acetylated α-tubulin (green) and the merged image with DAPI (blue) are shown. No fluorescence signal was detected in flagella in the control sperm treated by nonimmune serum from mouse. Typical images from at least three experiments are shown. Scale bar: 20 µm.

**Figure 6 ijms-23-01648-f006:**
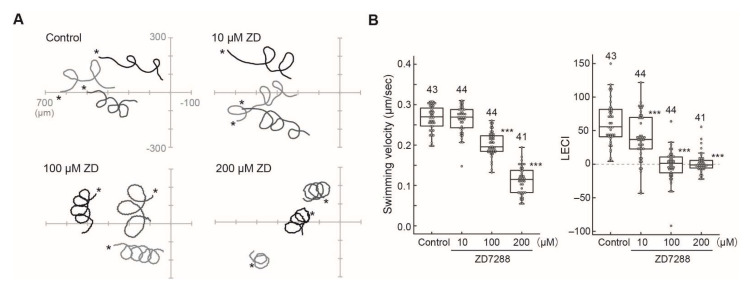
Effects of a HCN channel inhibitor ZD7288 on the chemotaxis of *Ciona* sperm. (**A**) Sperm swimming trajectories of four representative sperm in the presence of several concentrations of ZD7288 are shown. * The asterisks show the starting points of sperm swimming trajectory. (**B**) Comparison of sperm swimming velocity (Left) and linear equation chemotaxis indices (LECI) (Right) in control and inhibitor-treated sperm. Distribution of values is plotted in a box plot. Total number of observed spermatozoa from three experiments is shown on the top of each bar. *** Significant at *p* < 0.001 (Dunnett’s test) as compared with the control.

**Figure 7 ijms-23-01648-f007:**
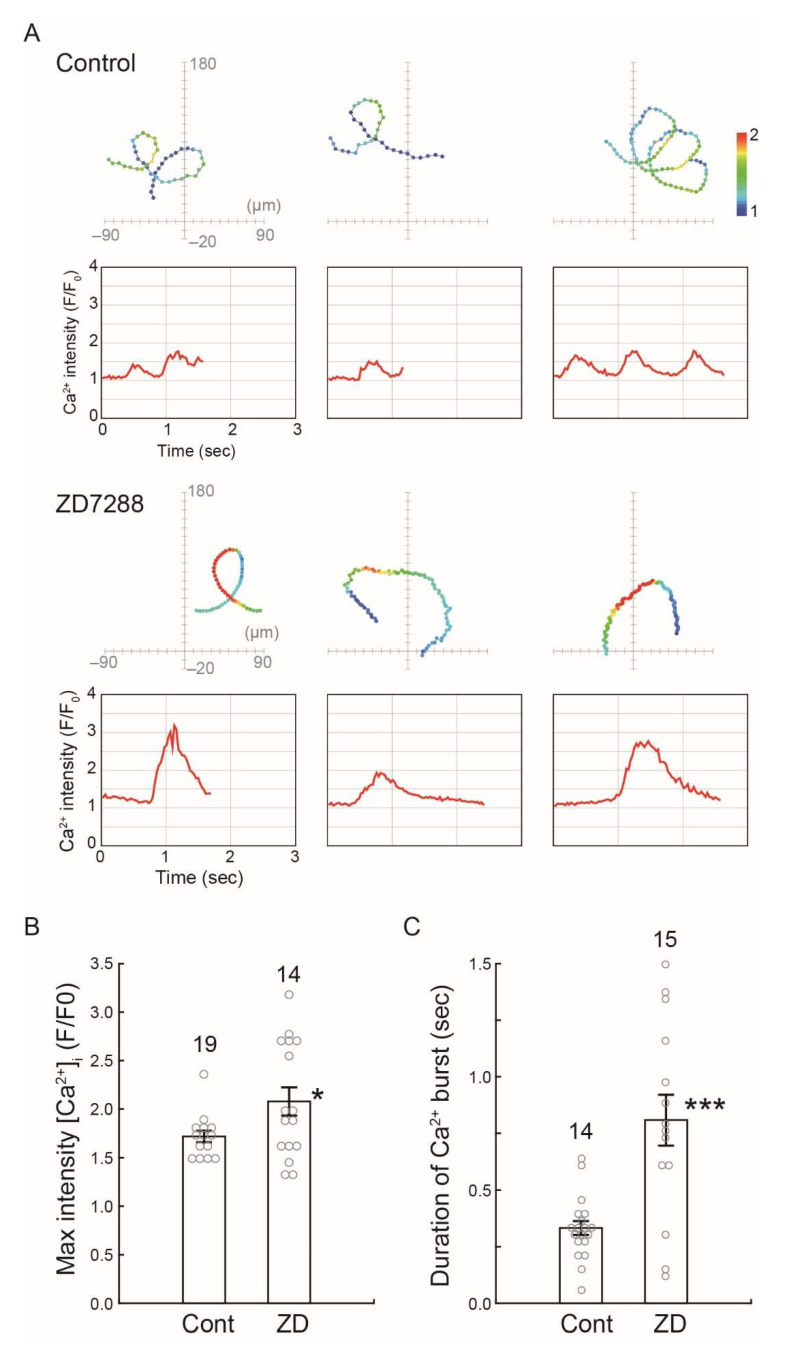
Effects of an HCN channel inhibitor ZD7288 on intracellular Ca^2+^ dynamics during sperm chemotaxis. (**A**) Trajectories of the sperm head (Upper) and changes in [Ca^2+^]_i_ signals yielded by the sperm tail (Lower) in ASW (control) or treated with 100 μM ZD7288 around the tip of a capillary containing 1 μM SAAF. The origin of the coordinates indicates the capillary tip. The arrows indicate the swimming direction of the sperm. The dots show the head position with the color representing the average intensity of [Ca^2+^]_i_ signals obtained from sperm tail in pseudocolors of the LUT. The color scale is the LUT for fluorescence signals. (**B**,**C**) Maximum of [Ca^2+^]_i_ (**B**) and duration of [Ca^2+^]_i_ increase (**C**) around the tip of a capillary containing 1 μM SAAF in sperm tail in ASW (control) or treated with 100 μM ZD7288. [Ca^2+^]_i_ is expressed as F/F0, which is the value of the fluorescent intensity from head (F) divided by the average intensity of basal [Ca^2+^]_i_ before SAAF stimulation emitted by the heads (F0). The values are expressed as mean ± S.D. Total number of observed spermatozoa from three experiments is shown on the top of each bar. * Significant at *p* < 0.05, *** *p* < 0.001 (Student’s *t*-test) as compared with the control.

## Data Availability

All data presented in this study are available in the article.

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
