# Peer review of "The Roles of Two CNG Channels in the Regulation of Ascidian Sperm Chemotaxis"

_ijms, 2022, doi:10.3390/ijms23031648_

Round 1

Reviewer 1 Report

This is an interesting study on the physiological role of CNG channels in ascidian sperm chemotaxis. The experimental design is appropriate, and the results obtained are of high scientific value.

Nevertheless, there are some considerations that authors must address before its acceptance for publication:

  1. Lines 234-239: Materials Please, indicate the number of animals included in the study, as well as the sperm concentration of semen samples collected from the sperm duct.
  2. Line 293. How many samples were analysed? And how many replicas per sample? Please, specify.
  3. Lines 314-315. Please, provide a brief description of the procedure followed. Indicate the number of samples and the number of replicas per sample analysed.
  4. Lines 326-329: Statistical analysis Did the authors check the normality and homoscedasticity of data obtained? If so, please indicate the test used to check it.
  5. A section highlighting the most relevant results obtained is lacking. Authors must provide a section/paragraph highlighting the biological relevance of the results obtained.

Reviewer 2 Report

Before further processing article that reports novel, interesting and unique data some cosiderations are listed

1.    Give concentation of used antibodies
2.    Provide controls in microphotographic doccumenation
3.    Microphotographs are to small
4.    Provide information on biology of Ciona intestinalis including reproductive physiology
5.    Provide information on used model Ciona intestinalis into title
6.    Provide information on number of samples used for studies

Round 2

Reviewer 1 Report

Authros made all changes requested by both reviewers. Nevertheless, a section/paragraph with the main conclusions of their work is still lacking.

Author Response

We thank the reviewer for a helpful comment.

We added a section of the conclusions to the main text. We also use the English editing services on MDPI and reflect it to the manuscript.